# Bacterial endosymbionts influence host sexuality and reveal reproductive genes of early divergent fungi

Stephen J. Mondo[1,2], Olga A. Lastovetsky[3], Maria L. Gaspar[4], Nicole H. Schwardt[1], Colin C. Barber[1], Robert Riley[2], Hui Sun[2], Igor V. Grigoriev [2,5] & Teresa E. Pawlowska [1]

Many heritable mutualisms, in which beneficial symbionts are transmitted vertically between host generations, originate as antagonisms with parasite dispersal constrained by the host. Only after the parasite gains control over its transmission is the symbiosis expected to transition from antagonism to mutualism. Here, we explore this prediction in the mutualism between the fungus *Rhizopus microsporus* (*Rm*, Mucoromycotina) and a beta-proteobacterium *Burkholderia*, which controls host asexual reproduction. We show that reproductive addiction of *Rm* to endobacteria extends to mating, and is mediated by the symbiont gaining transcriptional control of the fungal *ras2* gene, which encodes a GTPase central to fungal reproductive development. We also discover candidate G-protein-coupled receptors for the perception of trisporic acids, mating pheromones unique to Mucoromycotina. Our results demonstrate that regulating host asexual proliferation and modifying its sexual reproduction are sufficient for the symbiont's control of its own transmission, needed for antagonism-to-mutualism transition in heritable symbioses. These properties establish the *Rm-Burkholderia* symbiosis as a powerful system for identifying reproductive genes in Mucoromycotina.

[1] School of Integrative Plant Science, Plant Pathology and Plant Microbe-Biology, Cornell University, Ithaca, NY 14853, USA. [2] US DOE Joint Genome Institute, Walnut Creek, CA 94598, USA. [3] Graduate Field of Microbiology, Cornell University, Ithaca, NY 14853, USA. [4] Department of Molecular Biology and Genetics, Cornell University, Ithaca, NY 14853, USA. [5] Department of Plant and Microbial Biology, University of California Berkeley, Berkeley, CA 94720, USA. Stephen J. Mondo and Olga A. Lastovetsky contributed equally to this work. Correspondence and requests for materials should be addressed to T.E.P. (email: tep8@cornell.edu)

Heritable mutualisms are a source of major evolutionary innovations[1]. However, their evolution remains elusive. Evolutionary theory suggests that many heritable mutualisms originate as antagonisms in which parasite dispersal is controlled by the host[2]. A transition to mutualism requires the parasite to dominate the coevolutionary race with the host by establishing control over its own transmission. However, few symbioses exist where this prediction can be explored. One such system is the mutualism between a soil fungus *Rhizopus microsporus* (*Rm*, Mucoromycotina) and a beta-proteobacterium *Burkholderia*, which controls asexual proliferation of its host[3]. Like many other Mucoromycotina, the *Rm* hosts of *Burkholderia* thrive as soil saprotrophs. They can cause food spoilage, infect plants[4], and act as opportunistic pathogens of immune-compromised humans[5]. While the evolutionary history of the *Rm*-*Burkholderia* symbiosis is uncertain, present-day antagonistic interactions of *Burkholderia* endobacteria with nonhost *Rm* isolates naturally free of endobacteria[6] suggest that it originated as an antagonism. In the *Rm*-*Burkholderia* mutualism, the partners can be separated, cultivated independently, and reassembled to form a functional symbiosis in which the endobacteria reside directly in the host cytoplasm[3]. *Burkholderia* cells are transmitted via sporangiospores, which are asexual propagules produced by the host[3]. Sporangiospores are generated continuously throughout colony growth in favorable environmental conditions, disseminate aerially, and germinate rapidly.

In addition to asexual propagation, fungi, like most other eukaryotes, engage in sexual reproduction. In Mucoromycotina, sex involves the union of gametangia, leading to the formation of a zygospore[7]. In heterothallic species, such as *Rm*, two compatible strains, sex *plus* (sexP) and sex *minus* (sexM), are required for mating to be successful[7]. Partner recognition and progression of mating are mediated by trisporic acids and their precursors[8, 9], which act as sex pheromones and are synthesized in a cooperative manner from intermediates provided by the complementary mating partner[10]. Due to their recalcitrance to genetic analysis

and manipulation, Mucoromycotina are one of the least explored major lineages of fungi, with only few reproductive genes characterized functionally thus far[11]. To test the hypothesis that the *Burkholderia* endobacteria control sexual reproduction of the *Rm* host and identify the control mechanism, we mated fungi that harbored endosymbionts or were cured of them, followed by transcriptional profiling and phylogenomic analyses utilizing the wealth of information on sexual reproduction in Dikarya, a lineage uniting Ascomycota and Basidiomycota. We discovered that endobacteria modify sexual reproduction of *Rm*, and generated insights into the reproductive biology of Mucoromycotina.

## Results

**Diversity and natural loss of endobacteria.** Previous observations indicated that only a single bacterial cell was present per sporangiospore in one of the *Rm* strains, CBS112285[3]. We found that bacterial loads in other strains differ, varying from on average three cells per sporangiospore in strain ATCC 52813 to four in ATCC 52814 (Supplementary Fig. 1). This observation suggests phenotypic diversity among host–symbiont pairings.

The role of endobacteria in asexual proliferation of *Rm* was evidenced by the loss of sporulation in mycelia treated with antibiotics that eradicate *Burkholderia*[3]. We observed that endobacteria and asexual reproduction can be lost spontaneously after as few as four generations of propagating the fungus via single spores, and after 12 generations of propagation via mycelial fragments (Supplementary Table 1). These patterns indicate that in nature, hosts can become naturally cleared of their endosymbionts, and the loss of sporulation can be attributed to endosymbiont absence.

**Not all zygospores are populated by endobacteria.** All asexual sporangiospores formed by *Rm* strains hosting endobacteria appear to harbor the symbiont. To test whether the same is true for sexually produced zygospores, we mated *Rm* strains ATCC 52813 sexP and ATCC 52814 sexM, which both naturally contain

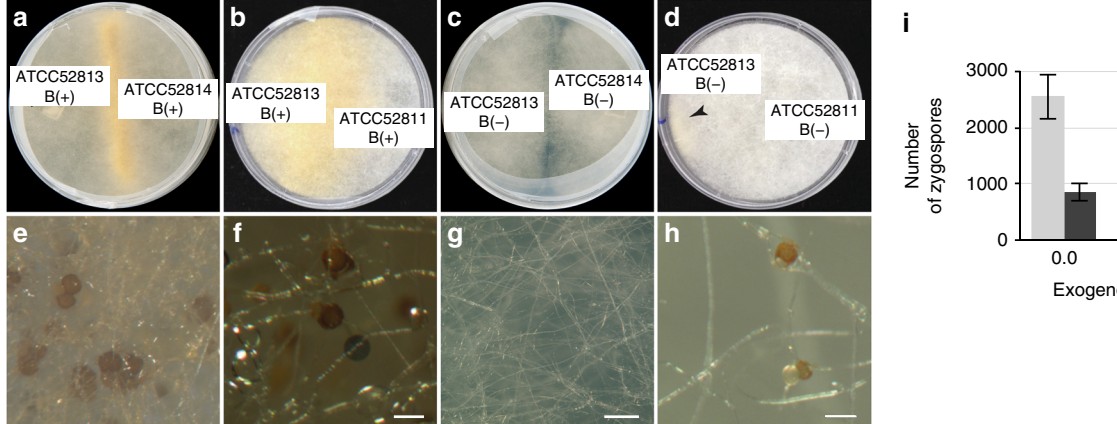

**Fig. 1** Impact of *Burkholderia* endobacteria on the reproductive biology of the *Rm* host. Successful mating between sex-compatible *Rm* B(+) strains: **a** ATCC 52813 and ATCC 52814, and **b** ATCC 52813 and ATCC 52811. **c** Complete loss of mating between B(−) isolates ATCC 52813 and ATCC 52814. **d** Restricted mating between B(−) isolates ATCC 52813 and ATCC 52811, with zygospores formed in the area indicated by an arrow. Accumulation of zygospores and β-carotene in the zone of interaction between B(+) mates: **e** ATCC 52813 and ATCC 52814, scale bar 100 μm, and **f** ATCC 52813 and ATCC 52811, scale bar 100 μm. **g** No sexual structures or β-carotene are apparent in the zone of interaction between B(−) mates ATCC 52813 and ATCC 52814 that show total loss of mating; scale bar 500 μm. **h** Rare zygospores produced during an interaction between B(−) mates ATCC 52813 and ATCC 52811 that resulted in restricted mating; scale bar 100 μm. **i** Effects of endobacteria and exogenous dibutyryl cAMP on the formation of zygospores during mating between B(+) ATCC 52813 and ATCC 52814 vs. mating between B(−) ATCC 52813 and ATCC 52814 showing limited zygospore formation. Increased concentrations of cAMP reduced the rate of zygospore formation in the interactions between B(−) mates that were capable of restricted mating (Student post hoc test of the interaction between bacterial presence and cAMP level in two-way ANOVA, $P = 0.02$), whereas the decrease in zygospore formation in interactions between B(+) mates was not statistically significant. Ten mating interactions were examined per condition. B(+), endobacteria present; B(−), endobacteria absent; error bars represent s.e.m.

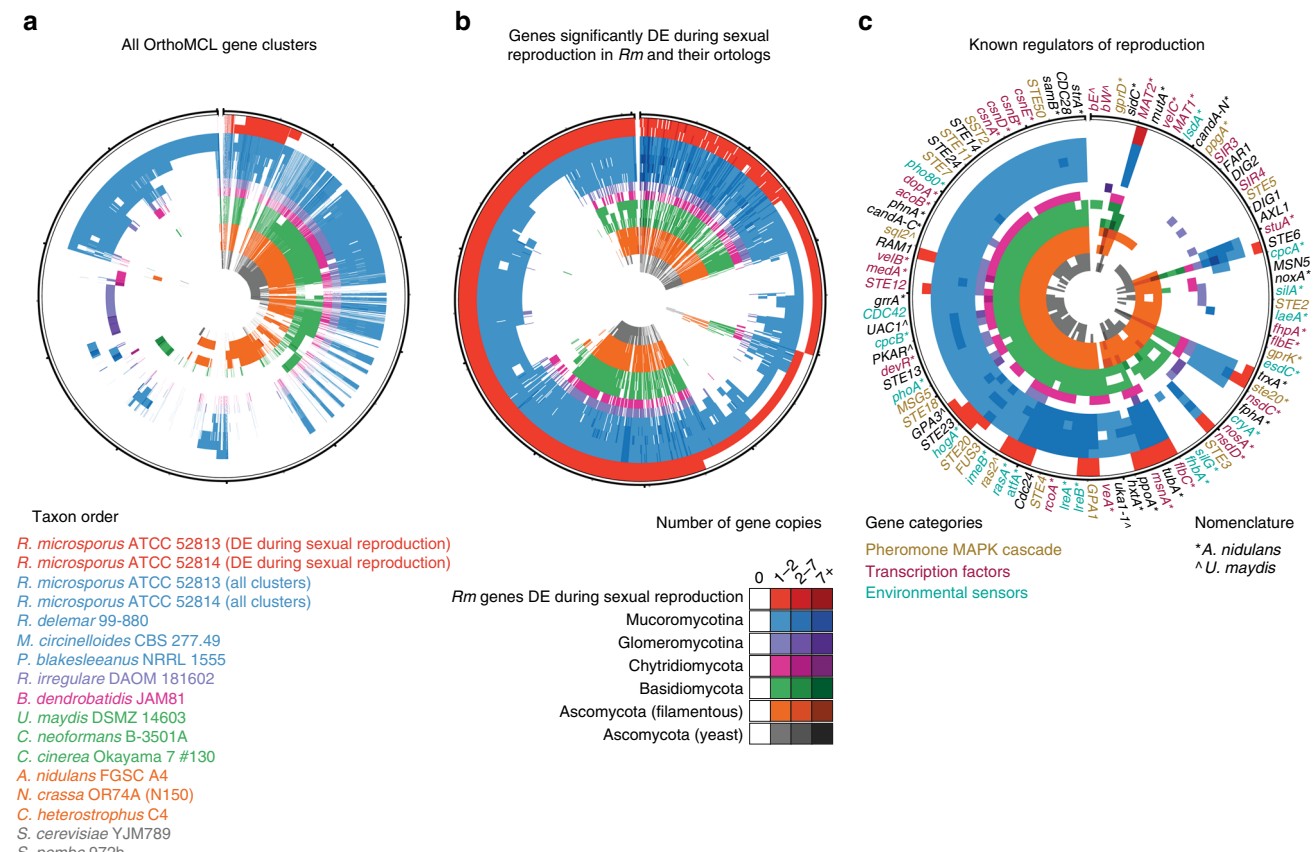

**Fig. 2** Many genes differentially expressed during *Rm* sexual reproduction are unique to Mucoromycotina. Tracks show gene content (copy number) for each individual genome, and each column represents an individual OrthoMCL cluster. Counts of gene copies within an individual cluster are shown in increasingly darker shades of color: red, DE genes in *Rm*; blue, Mucoromycotina; purple, Glomeromycotina; pink, Chytridiomycota; green, Basidiomycota; orange, filamentous Ascomycota; gray, Ascomycota yeasts. **a** All OrthoMCL gene clusters. **b** Genes DE during sexual reproduction of *Rm*. **c** Regulators of sexual reproduction. Gene names are colored based on what aspect of sexual reproduction they are involved in: mustard, pheromone MAPK cascade; purple, transcription factors; teal, environmental sensors. All data are presented in Supplementary Data 2, and are ordered based on Fig. 2a

endobacteria in their mycelia. We then surveyed zygospores for bacterial presence by PCR targeting their 23S rRNA gene. *Burkholderia* was detected in 40% (±6% s.e.m.) of zygospores, suggesting that the rate of symbiont transmission through the sexual pathway is substantially lower than through the asexual pathway.

**Endobacteria modify fungal mating**. To test the hypothesis that *Burkholderia* endobacteria control sexual reproduction of *Rm*, we examined the interactions of wild-type B(+) strains that harbored endobacteria and B(−) isolates that were cured of endosymbionts: (1) ATCC 52813 sexP with ATCC 52814 sexM, (2) ATCC 52813 sexP with ATCC 52811 sexM, and (3) ATCC 62417 sexP with ATCC 52811 sexM. We found that in all pairs, bacteria impacted fungal ability to reproduce sexually. They either controlled it completely, with removal of endobacteria leading to total loss of mating, or incompletely, with removal of endobacteria leading to a reduced zygospore yield (Fig. 1). These two outcomes did not appear to be specific to the strains that were mated. Moreover, loss of mating was not a consequence of vigor reduction in cured isolates, as, with the exception of ATCC 62417, which grew poorly after the loss of endobacteria, the differences in the rate of mycelial expansion between the B(−) and B(+) mycelia were largely negligible (Supplementary Fig. 2).

**Attempts to restore fungal mating**. To examine whether the loss of host reproduction upon removal of endobacteria was reversible, we investigated the consequences of reintroducing

endosymbionts into the cured host isolates that exhibited total loss of mating. Reinfection with endobacteria restored both asexual and sexual reproduction regardless of whether bacteria were introduced into their original native hosts or nonnative hosts, with all pairwise combinations of source bacteria and target fungal strains across ATCC 52811, ATCC 52813, ATCC 52814, and ATCC 62417 yielding reproduction. However, while asexual reproduction was restored immediately after reinfection, restitution of the ability to mate required an additional step of exposing the reinfected isolates to extreme cold (−80 °C), which eliminated fungal hyphae while preserving sporangiospores. This observation suggested that there was a subtle mechanistic difference in how endosymbionts interact with asexual vs. sexual reproduction of the fungus. We also attempted to restore *Rm* sexual reproduction in the absence of endosymbionts by exposing compatible B(−) mates to extracts of mated mycelia and to environmental conditions known to affect reproduction in Mucoromycotina (Supplementary Table 2). None of these treatments restored mating or asexual propagation in cured fungi, suggesting that for reproduction, the host is addicted to bacterial factors.

**Gene networks responsible for mating in Mucoromycotina**. To investigate the genetic underpinnings of fungal reproductive responses to endobacteria, we conducted an RNA-seq experiment assessing global gene expression patterns under six different conditions: (1) *Rm* ATCC 52813 B(+) grown alone, (2) ATCC 52813 B(−) grown alone, (3) ATCC 52814 B(+) grown alone, (4)

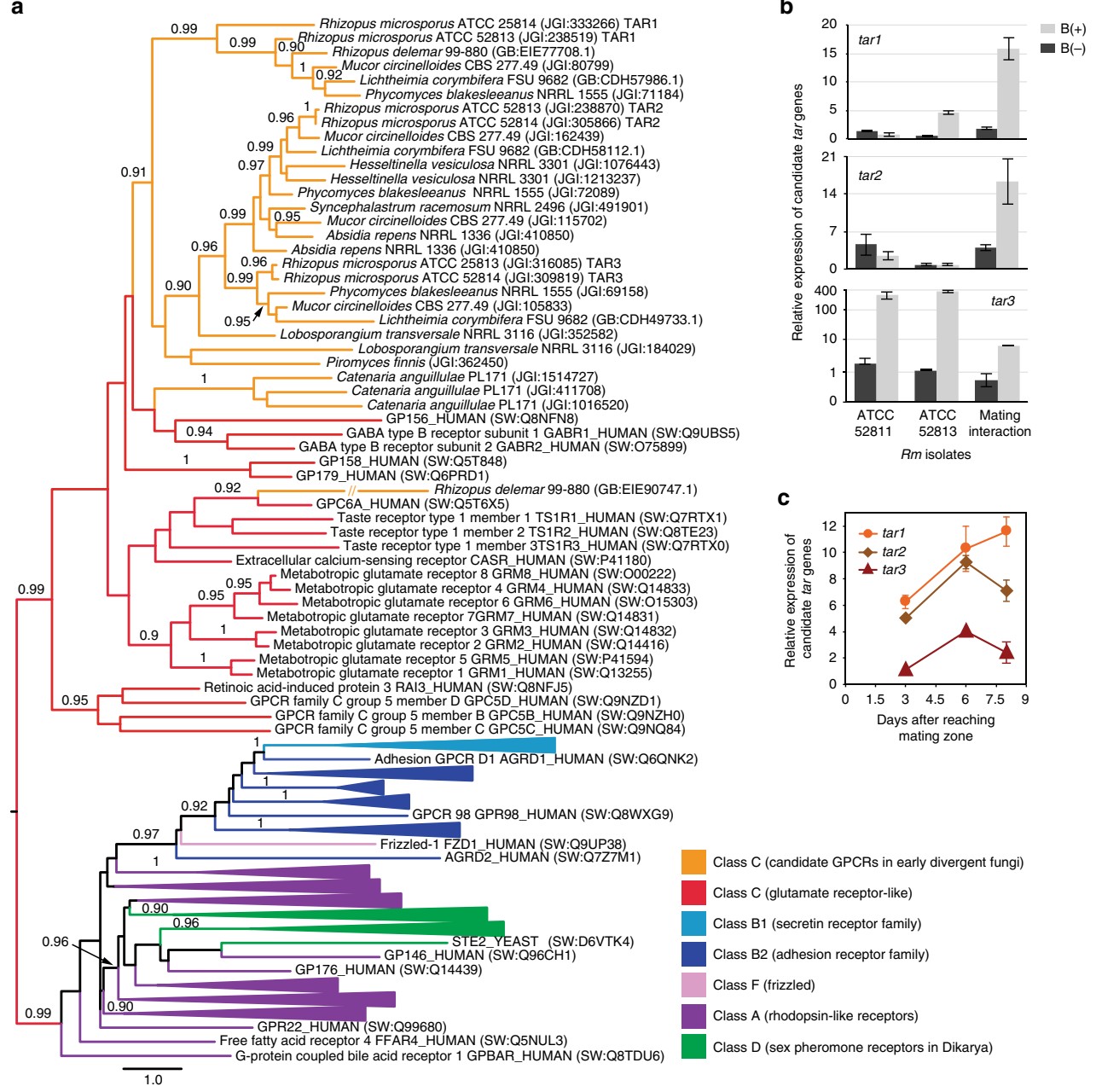

**Fig. 3** Candidate trisporic acid receptors are class C GPCRs. **a** Maximum-likelihood phylogeny of seven-transmembrane domains from early-diverging fungi (orange) and five different classes of G-protein-coupled receptors (GPCRs): A (purple), B1 (light blue), B2 (dark blue), F (pink), as well as the fungal-specific class D pheromone receptors (green). Fungal protein identifiers include species and strain designation, as well as GenBank accession number or JGI protein ID. Nonfungal proteins have GPCRdb[74] identifiers and UniProtKB[75] (SW) accession numbers. Support values are displayed above branches. The GPCR sequence alignment and the complete phylogeny are included in Supplementary Data 3 and Supplementary Data 4, respectively. TAR, candidate trisporic acid receptor. **b** Expression levels of the candidate *tar* genes in *Rm*. Three biological replicates were examined per condition. B(+), endobacteria present; B(−), endobacteria absent; error bars represent s.e.m. **c** Changes of the candidate *tar* gene expression patterns during the progression of mating in *Rm* B(+) mates measured in three replicate cultures per time point. Error bars represent s.e.m.

ATCC 52814 B(−) grown alone, (5) both B(+) mates grown together, and (6) both B(−) mates grown together, with sexual reproduction completely absent. We sequenced the genomes of both mates (Supplementary Tables 3 and 4, and Lastovetsky et al.[6]) to facilitate mapping of RNA-seq reads to each host. Because the knowledge of molecular mechanisms underlying sexual reproduction in Mucoromycotina lags behind other fungi, we first needed to identify genes that are relevant to reproductive processes in these fungi.

We found that in *Rm*, 2124 genes were differentially expressed (DE) at a significant level during sexual reproduction relative to asexual growth and sporulation, with 1496 genes upregulated and 628 downregulated (false-discovery rate (FDR) corrected $P \leq 0.05$, identified using DESeq[12]; Supplementary Data 1, Supplementary Fig. 3). Ortholog clustering revealed that, while many of these genes were conserved across most fungi (57.95%), a substantial proportion were either restricted to Mucoromycotina (42.04%) or even unique to the *Rm* lineage (11.67%; Fig. 2, Supplementary

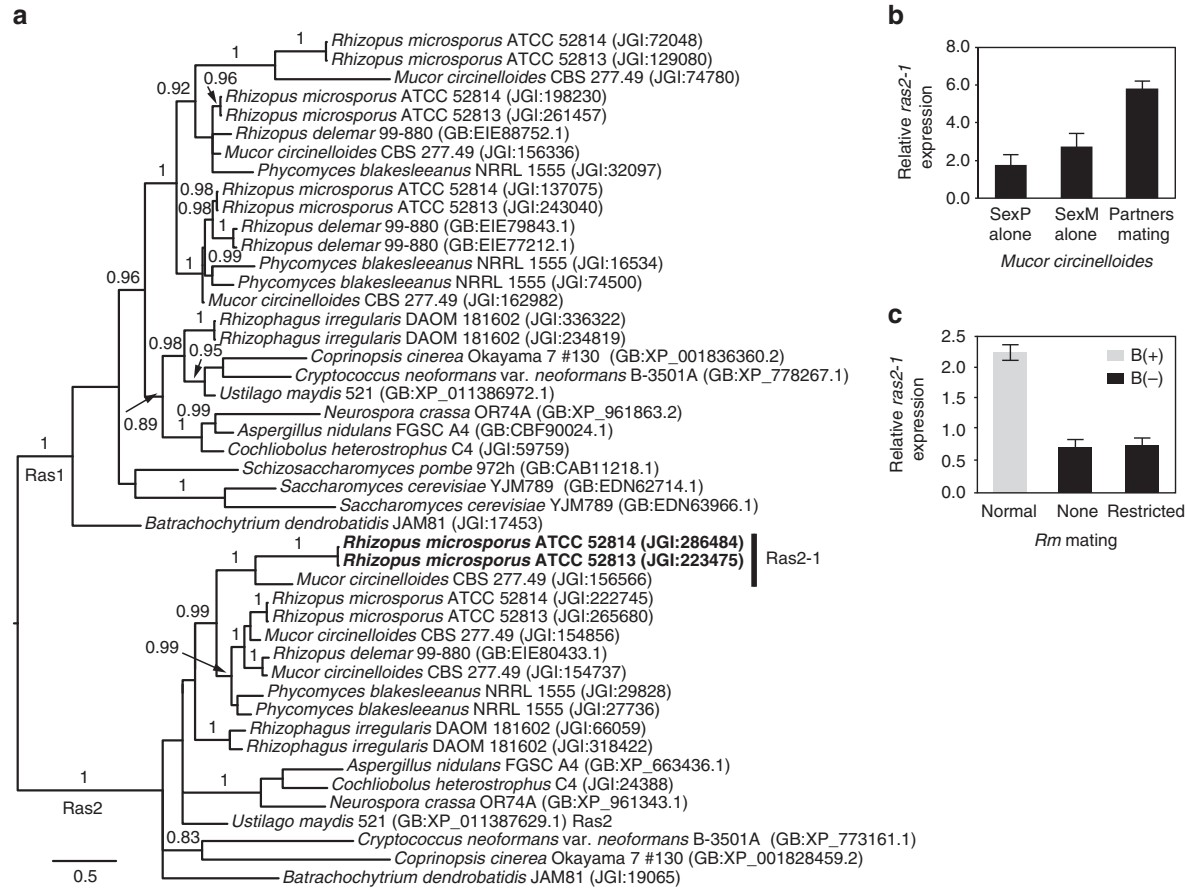

**Fig. 4** Ras proteins in Mucoromycotina. **a** Bayesian phylogeny of proteins in the Ras1 and Ras2 ortholog groups. Protein identifiers include species and strain designation, as well as GenBank accession number or JGI protein ID. The *Rm ras2* genes DE due to endobacteria and mating are in boldface. Posterior probability values are displayed above branches. The alignment is included in Supplementary Data 5. **b** Expression levels of *ras2-1* in strains of *Mucor circinelloides* ATCC 1216b sexP and CBS 277.49 sexM measured in mates grown alone before the onset of asexual sporulation and during a mating interaction. Three replicate cultures were examined per condition. Error bars represent s.e.m.. **c** Expression levels of the *ras2-1* gene during mating between *Rm* ATCC 52813 and ATCC 52811 exhibiting normal mating, restricted mating, and total loss of mating. Each condition had three biological replicates. B(+), endobacteria present; B(−), endobacteria absent; error bars represent s.e.m.

Table 5, Supplementary Data 2). For example, several major regulators of sexual reproduction, such as genes interacting with the pheromone mitogen-activated protein kinase (MAPK) cascade and several transcription factors appear to consistently play roles in sexual reproduction across different fungi, including Mucoromycotina (Supplementary Data 2). Conversely, genes encoding response regulators, such as PHOA/Pho80[13] (phosphate response) and CpcB[14] (amino acid starvation), which provide important cues for sexual reproduction in *Aspergillus nidulans* but are not involved in sex regulation of other Dikarya, did not show differential expression during sexual reproduction in *Rm* (Supplementary Table 5, Supplementary Data 1). As expected, we also observed upregulation of mating-type genes, *sexM* (FDR corrected $P = 1.08 \times 10^{-22}$) and *sexP* (FDR corrected $P = 9.61 \times 10^{-188}$), previously characterized in Mucoromycotina[11]. In contrast, the neighboring *rhnA* gene[15] was not differentially expressed during reproduction in *Rm*. Finally, we discovered that Mucoromycotina lack all the mating pheromone recognition genes present in Dikarya (Fig. 2; Supplementary Data 2), a pattern consistent with the fact that Mucoromycotina use trisporic acids and their precursors as sex pheromones rather than the peptides and lipopeptides employed by Dikarya[16].

**Candidate trisporic acid receptors.** As G-protein-coupled receptors (GPCRs) are involved in the perception of mating

pheromones in other fungi[17] and animals[18], we explored *Rm* genes upregulated during sexual reproduction in search of GPCRs unique to the Mucoromycotina lineage. We found three such candidate trisporic acid receptor (*tar*) genes, encoding class C seven-transmembrane domain GPCRs, that were present in the genomes of ATCC 52813 and ATCC 52814 (Fig. 3a). Through qRT-PCR, we confirmed that expression of these three candidate genes was also upregulated during mating interactions of a different pair of *Rm* mates, ATCC 52813 and ATCC 52811 (Fig. 3b). In addition, we established that the expression pattern of each gene changes during the progression of mating (Fig. 3c). Through phylogeny reconstructions, we determined that these GPCRs cluster with animal γ-aminobutyric acid (GABA) receptors, a group related to animal retinoic acid-inducible class C GPCRs (Fig. 3a).

**Effects of endobacteria on regulation of *Rm* mating.** Only 80 of the 2124 *Rm* sex-related transcripts were impacted by the symbiont presence in the hosts (Supplementary Fig. 3, Supplementary Data 1). One of the most striking observations was an ~12-fold downregulation in the absence of endobacteria of an ortholog of *ras2*, which encodes a small GTPase protein. Furthermore, *ras2* was downregulated in fungi cured of bacteria growing alone vegetatively and upregulated during active asexual proliferation. Despite the expansion of the *ras* gene family in Mucoromycotina

relative to Dikarya (Fig. 4a), only the expression of the *Rm ras2-1* gene (protein ID 223475 and 286484 in *Rm* ATCC 52813 and ATCC 52814, respectively) was affected by endobacteria. Ras2 is conserved across all fungi except Ascomycota yeasts (Fig. 4a). In the basidiomycete *Ustilago maydis*, it controls the initiation of the mating pheromone MAPK cascade[19]. Ras2 is also involved in asexual reproduction, as shown in ascomycetes in which repression of *ras2* has a negative impact on sporulation[20, 21]. To examine whether the role of *ras2-1* extends to mating in other Mucoromycotina, we conducted qRT-PCR on vegetative mycelia vs. mating interactions involving ATCC 1216b sexP and CBS 277.49 sexM strains of *Mucor circinelloides*, and found that expression of *ras2-1* was elevated during mating relative to vegetative growth (Fig. 4b). Finally, we examined expression levels of *ras2-1* in mating interactions between *Rm* ATCC 52813 and ATCC 52811 focusing on patterns exhibited by cured mates that experience total loss of mating vs. restricted mating. Like in the RNA-seq experiment, the levels of *ras2* expression were lower in interactions between B(−) mates compared to those between B(+) mates (Fig. 4c). However, they did not differ between interactions of B(−) mates exhibiting total loss of mating vs. restricted mating (Fig. 4c), a pattern consistent with incomplete control of mating by the endobacteria.

In addition to regulating the pheromone MAPK cascade, in *U. maydis*, Ras2 interacts with the cyclic adenosine monophosphate (cAMP) signaling pathway and controls morphogenesis[19]. cAMP is a secondary messenger that, in coordination with the pheromone MAPK cascade, affects sexual development in many fungi, albeit often with contrasting effects[22]. We explored the impact of exogenous cAMP on *Rm* mating by exposing B(+) and B(−) mates to 0 mM, 1 mM, and 2 mM dibutyryl cAMP. We found that increased concentrations of cAMP reduced the rate of zygospore formation in the interactions between B(−) mates that were capable of restricted mating (Student post hoc test of the interaction between bacterial presence and cAMP level in two-way ANOVA, $P = 0.02$), whereas the decrease in zygospore formation in interactions between B(+) mates was not statistically significant (Fig. 1i). These results suggested that elevated levels of cAMP interfered with mating in *Rm*, and the endosymbiont presence buffered the negative effects of high cAMP levels on sexual reproduction.

## Discussion

We found that *Rm* is highly dependent for survival on the *Burkholderia* endobacteria. Although the fungus can grow vegetatively after endobacteria are lost naturally or eradicated with antibiotics[23], it is unable to proliferate asexually and its ability to reproduce sexually is severely compromised. These patterns indicate that in the *Rm*-*Burkholderia* mutualism, the endosymbiont controls its own vertical transmission, which is a prerequisite for the antagonism-to-mutualism transition in heritable symbioses[2]. Remarkably, less than half of zygospores formed during mating interactions of wild hosts harbor endobacteria. However, as zygospore germination is extremely difficult to achieve under laboratory conditions[24], it remains untested whether zygospore functionality is affected by endosymbiont absence. Zygospore germination occurs via a sporangium resembling the asexual sporangium. Therefore, it cannot be excluded that endobacteria-free zygospores may fail to germinate, further reinforcing endosymbiont dominance over host reproduction. Endosymbiont ability to control its own transmission is expected to facilitate reciprocal selection between the partners, leading to utilization of symbiont services by the host[2]. In the *Rm*-*Burkholderia* symbiosis, these services include endosymbiont-mediated synthesis of rhizoxin, a potent toxin that enables pathogenesis of plants by *Rm*[23, 25].

In addition to facilitating the antagonism-to-mutualism transition, the role of endobacteria in regulating *Rm* asexual and sexual reproduction is consistent with the addiction model of mutualism evolution[6, 26]. According to this model, a host population that interacts with an antagonistic symbiont should develop mechanisms to compensate for its negative effects and become addicted to the symbiont's continued presence[27]. In the case of the *Rm*-*Burkholderia* symbiosis, endobacteria have hijacked an indispensable component of the host's developmental machinery by gaining control over expression of *ras2-1*, encoding a G-protein responsible for the reproductive development in Dikarya[19–21]. The exact mechanism of bacterial control over the *ras2-1* expression and the evolutionary trajectory that lead to it are unknown. However, as stimulation of *ras* signaling induces programmed cell death in other fungi[28], it is possible that in the ancestrally antagonistic relationship between *Rm* and *Burkholderia*[6], establishing control over the *ras2-1* expression by endosymbionts was an important component of coevolution between partners, leading to adaptive changes in host regulation of *ras2-1* signaling (Supplementary Fig. 4). Evolutionary theory predicts further that once a mutualism is established, the host is favored to control mixing of symbionts[29]. Such control is expected to reduce harmful competition among symbionts for the host resources. It remains to be investigated whether endobacteria are able to mix in the *Rm*-*Burkholderia* symbiosis.

We exploited the endosymbiont control over *Rm* reproduction to reconstruct the key reproductive pathways across the fungal kingdom, including Mucoromycotina, Ascomycota, and Basidiomycota. Since many of these genes were experimentally studied in Dikarya, we were able to augment our findings with information on conservation of their sex-related function. Using this approach, we uncovered candidate genes that may be involved in perception of trisporic acid pheromones in Mucoromycotina. Unlike class D GPCRs responsible for pheromone sensing in Dikarya[30], which are absent from *Rm*, these candidate receptors appear to represent class C GPCRs. They are encoded by genes that are conserved across all Mucoromycotina, upregulated during sexual reproduction in *R. microsporus*, absent from higher fungi, and closely related to retinoic acid-sensing GPCRs in animal systems. Similar to trisporic acid, retinoic acid is derived from β-carotene and is essential for the initiation of meiosis in animals[31, 32]. Further functional analyses are now required to test the hypothesis that these C GPCRs interact with and transduce trisporic acid pheromone signals.

Overall, our findings indicate that in the *Rm*-*Burkholderia* symbiosis, endobacteria control their vertical transmission by regulating asexual proliferation of the host and impacting its mating. Such control appears to be sufficient to have mediated the antagonism-to-mutualism transition in this heritable symbiosis. Symbiont presence correlated with expression levels of the host *ras2-1* gene, a major regulator of both sexual and asexual reproduction in fungi, suggesting that endobacteria influence its activity. Finally, we took advantage of the symbiont impact on host mating to make inferences about reproductive genes in Mucoromycotina, a group of fungi recalcitrant to genetic analysis. In the process, we discovered candidate trisporic acid receptors, TARs, that may be responsible for perception of trisporic acid sex pheromones uniquely utilized by this group of fungi.

## Methods

**Rm strains, culture conditions, and loss of endobacteria**. Strains ATCC 52811, ATCC 52813, ATCC 52814, and ATCC 62417 were cultivated on half (1/2) or full-strength potato dextrose agar (PDA) containing 2 g L⁻¹ potato extract, 10 g L⁻¹ dextrose, and 15 g L⁻¹ agar. Plates were sealed with Parafilm M (Pechiney Plastic

Packaging Company), unless otherwise noted. In addition to curing fungi of *Burkholderia* endobacteria with antibiotics as described in Partida-Martinez and Hertweck[23], we experimented with the impact of their subculturing by individual spores and mycelial fragments on bacterial presence. Isolates of ATCC 52813 were propagated by mycelial fragments and by single spores, with two replicates per isolate. For mycelial fragments, a small spore-free section of ~1 × 0.5 cm was excised from the edge of the colony and transferred to fresh PDA. For single-spore propagation, sporulating colonies were flushed with 3 mL of sterile nanopure water to dislodge spores. To remove hyphae, the spore suspension was filtered through a cotton filter into a 2-mL microfuge tube, and diluted $10^{-3}$ in sterile nanopure water, followed by spreading 100 μL into a 6-cm PDA plate. After approximately 24 hours of incubation at 30 °C, one colony was removed and plated onto a PDA plate. The absence of endosymbionts was confirmed by PCR using *Burkholderia*-specific primers GlomGiGf[33] and LSUb 483r[34] as described in Mondo et al[34].

**Extraction, cultivation, and reintroduction of endobacteria.** Young fungal mycelium (1–2-days old) containing endosymbionts was finely chopped in 500-μL Luria-Bertani (LB) broth, and pressed gently to release cellular contents into the broth, followed by filtration through a 2-μm filter to remove fungal debris. Varying amounts of filtrate were added to LB plates containing 10 mL L$^{-1}$ glycerol and 100 μg mL$^{-1}$ cycloheximide. Single colonies were isolated and grown at 30 °C either on LB agar or in 5-mL LB broth incubated at 250 rpm. To reinfect fungi with endobacteria, a plug of agar was removed from 1/2 PDA using the upper end of a P-1000 pipette tip and replaced with a plug of LB agar. Bacterial inoculum was placed on the LB agar plug, and a plug containing cured fungus was either positioned directly on bacterial cells, or somewhere nearby on the plate.

**Visualization of endobacteria in sporangiospores.** For visualization, endobacteria were transformed with either an mCherry-expressing or YFP-expressing gentamicin resistance-conferring plasmid pBS46, and introduced to cured strains ATCC 52813 and ATCC 52814. Sporangiospores were collected and embedded in polyacrylamide pads to immobilize them[35]. Between 70 and 100 spores were examined using the DeltaVision RT system (Applied Precision) with an Olympus IX-70 inverted microscope (Olympus America) fitted with a 100-W mercury arc bulb and a CoolsnapHQ ICX285 camera (Sony). Z-sections were acquired using 0.15-μm step size and the Z stacks were deconvolved using the softWoRx Explorer software (Applied Precision).

***Rm* mating interactions.** Half-strength PDA plates were used for all mating experiments. Mates were placed at the edges of the plate, allowing mycelia to grow toward each other and develop an interaction zone in the center of the plate. Cultures were incubated in the dark at 30 °C. Each interaction was examined in 6–10 replicates; the entire experiment, including removal of endobacteria, was repeated three times. Prior to investigating the role of endobacteria in sexual reproduction, we examined fungal mating in both darkness and in ambient light, and found that *Rm* has no light preference with respect to mating. To ensure that loss of fertility in cured fungi is not due to multiple rounds of subcultivation[7] required to remove endobacteria[23], we generated tester lines of *Rm*-harboring bacteria that were subjected to the same number of subcultivation events as cured lines but without an antibiotic. We found that subculturing had no effect on sexual reproduction.

**Detection of endobacteria in *Rm* zygospores.** Mating interactions between *Rm* ATCC 52813 and ATCC 52814 were conducted on PDA (Sigma) as described above, and zygospores were analyzed after 7 days of incubation in the dark at 30 °C. A tuft of mycelium containing zygospores was removed from the mating zone and placed in 10% w/v chloramine T (Sigma) for 20 min to kill hyphae. The tuft was subsequently transferred to sterile water and shaken slowly for 5 min at room temperature, followed by two additional water washes lasting for 20 min. The mycelial tuft was then transferred onto a sterile 1.5% water agar and zygospores were collected with sterile forceps, taking care to remove all attached hyphae. A total of 80 zygospores from two separate mating interactions were transferred individually into 0.2-mL PCR tubes, crushed, and subjected to whole-genome amplification (WGA) with the Illustra GenomiPhi Kit v2 (GE) to generate template DNA for multiple PCR reactions per zygospore. A volume of 1 μL of the last water wash was used as a negative control during WGA and subsequent PCR, with a total of 16 negative controls. PCR was performed on the 1/20-diluted WGA products using LR1 and NDL22[36] as well as *Burkholderia*-specific primers[34] to detect the presence of fungal and bacterial DNA in zygospores.

**Attempts to restore mating in the absence of endobacteria.** Interactions between compatible B(−) mates ATCC 52813 and ATCC 52814 showing total loss of mating were examined on 1/2 PDA: (1) amended with mating relevant compounds, such as 3 g L$^{-1}$ NaNO$_3$, 0.5 g L$^{-1}$ NaNO$_2$, 1 mg L$^{-1}$ thiamine, and 16.6 mg L$^{-1}$ β-carotene as well as with extracts from successfully mating liquid cultures filtered through 0.22-μm filter, (2) at high moisture generated by adding 1-mL H$_2$O to culture plates and sealing them with Parafilm M, and at low moisture generated by sealing plates with porous Micropore™ surgical tape (3 M Health Care), (3) in ambient light and in darkness, (4) after cold treatment of 1-week, 2-week, 4-week,

and 10-week incubation at 4 °C, (5) under oxidative stress of 0.3, 0.6, 1.2, 2.4, and 10 mM H$_2$O$_2$, and (6) osmotic stress of 0.4, 0.8, and 1.6 M NaCl. Each treatment included three to five replicates. Exogenous dibutyryl cAMP (Enzo Life Sciences) was added at concentrations of 0 mM, 1 mM, and 2 mM to B(+) mates ATCC 52813 and ATCC 52814 and to B(−) mates ATCC 52813 and ATCC 52814 that showed restricted levels of mating. Ten mating interactions were established per condition and incubated in the dark at 30 °C for 10 days.

**RNA-seq.** Six different conditions were examined in *Rm* ATCC 52813 and ATCC 52814: B(+) and B(−) mates grown alone, as well as both B(+) and B(−) partners grown together, respectively. We chose to analyze the interaction between B(−) mates that exhibited total loss of mating to maximize our chances of identifying genes impacted by bacteria during sexual reproduction. For each condition, fungal plugs were placed at the edge of the 1/2 PDA plate and harvested after six days, when opposite B(+) mates were undergoing sexual reproduction. Each condition had two biological replicates, each consisting of five culture plates, which were pooled prior to RNA extraction. Total RNA was extracted from a 2.5-cm-wide strip of mycelium from the middle of the plate where most mating occurred, using the Ambion ToTALLY Total RNA Isolation Kit (Life Technologies) to recover both fungal and bacterial transcripts. Fungal rRNA was removed with the Human/Mouse/Rat Ribo-Zero rRNA Removal Kit (Epicentre), whereas bacterial rRNA was treated with the Gram-negative bacteria Ribo-Zero rRNA Removal Kit (Epicentre). After rRNA removal, sequencing libraries were constructed using the TruSeq RNA Sample Preparation Kit (Illumina) and sequenced at the Cornell University Biotechnology Resource Center using the Illumina Hi-Seq 100-bp paired-end platform.

Illumina reads were quality controlled using BBDuk (https://sourceforge.net/projects/bbmap/), and then mapped to either the *Rm* ATCC 52813 or ATCC 52814 genomes (depending on the sample) using TopHat2[37]. HTSeq[38] was used to collect read counts per gene, followed by exploration of differential expression across several comparisons using DESeq[12]: (1) ATCC 52813 B(+) vs. ATCC 52813 B(−), (2) ATCC 52814 B(+) vs. ATCC 52814 B(−), (3) both B(+) mates grown together (active mating) vs. both B(−) mates grown together (no mating), and (4) both B(+) mates grown alone vs. both B(+) mates grown together (active mating). This combination of comparisons allowed us to identify genes that were significantly DE in both mates due to (1) endosymbiont presence and (2) sexual reproduction, as well as assess the overlap between them. For comparisons involving both mates, we mapped reads to a combined assembly of both genomes, and since these strains are so closely related, we allowed reads mapped twice to be retained for downstream DESeq analysis. Genes with an adjusted $P \leq 0.05$ were considered DE.

**qRT-PCR.** Mating interactions were set up as above between *Rm* ATCC 52813 sexP and ATCC 52811 sexM. Expression of *ras2-1* was examined in (1) B(+) mating cultures, (2) cultures of B(−) mates exhibiting complete loss of mating, and (3) cultures of B(−) mates exhibiting restricted levels of mating. Expression of candidate *tar1*, *tar2*, and *tar3* genes was measured in B(+) and B(−) mates grown alone, as well as during mating interactions, with a particular focus on B(+) mating cultures at 3, 6, and 8 days after the mates reached the mating zone. To assess the expression of *ras2-1* in *Mucor circinelloides* (*Mc*), ATCC 1216b sexP and CBS 277.49 sexM were grown at 20 °C on YXT medium containing 4 g L$^{-1}$ yeast extract, 10 g L$^{-1}$ malt extract, 4 g L$^{-1}$ glucose, and 15 g L$^{-1}$ agar, with the pH adjusted to 6.5. *Mc* mates were grown (1) individually for 3 days and harvested before the onset of asexual sporulation, and (2) together and harvested 6 days after the onset of mating. Each condition had three biological replicates, each replicate consisting of a single plate. RNA was extracted with the Ambion ToTALLY Total RNA Isolation Kit (Life Technologies) and converted to cDNA using ProtoScript II First Strand cDNA Synthesis Kit (New England Biolabs). Real-time PCR was performed on a StepOnePlus™ Real-Time PCR System (Applied Biosystems) using TaqMan® Universal PCR Master Mix. The reaction mix consisted of 0.5 μM primers, 0.2 μM TaqMan® probe, 1× Master Mix, and 10 ng of cDNA in a volume of 25 μL. All reactions were performed in technical duplicate. Nontemplate control (10 ng of RNA) and nonreaction control (RNase-free water) were included. The thermal program for the PCR consisted of Stage 1: 95 °C, 10 min; Stage 2: 95 °C, 0.5 min and 60 °C, 1 min for a total of 40 cycles; and Stage 3: hold at 4 °C. Relative quantitation was conducted using the ΔΔC*t* method (StepOnePlus™ user manual of Applied Biosystems). Each RT-PCR experiment was performed in triplicate. Probes and primers were *Rm ras2-1* TaqMan® probe 5′-[6-FAM] AG CAT TTT ACT CAG TTG CT [Tamra-Q]-3′, *Rm ras2-1* forward primer 5′-CGC AAA GAC TTG TGC TAA TGT AGA A-3′, *Rm ras2-1* reverse primer 5′-CGG GCT TGC TTG ATC TGA-3′, *Rm tar1* (238519) TaqMan® probe 5′[6-Fam] ATT GCA AGA CTT GGC TAG T [Tamra-Q]3′, *Rm tar1* forward primer 5′-GGT GGC CGG GAA AAG G-3′, *Rm tar1* reverse primer 5′-TAG GTG TCA TCG AAC TCG TGT TAA A-3′, *Rm tar2* (238870) TaqMan® probe 5′[6-FAM] AGA GTG GAG CAC GGA T [Tamra-Q]3′, *Rm tar2* forward primer 5′-TCT GAA TTC GGC ACT GAC AAA CT-3′, *Rm tar2* reverse primer 5′-GAT TCG CTG CGA CCA TGA T-3′, *Rm tar3* (316085) TaqMan® probe 5′[6-Fam] CCT CTT CTT GGA CCT CT [Tamra-Q]3′, *Rm tar3* forward primer 5′-CGT CGG CAT TAT CGG AGA TAT C-3′, *Rm tar3* reverse primer 5′-TGC AAG GTG CTC GTC ATC A-3′, *Mc ras2-1* (156566) TaqMan® probe 5′[6-FAM] CTC GCA CGC TTA AT [Tamra-Q]3′, *Mc ras2-1* forward primer 5′-TCG AGA AGA GGG AGC GAT AAA G-3′, and *Mc ras2-1*

reverse primer 5′-TGC TGA CGT CTC GAC GAA AT-3′. *Rm* ATCC 52813 gene encoding hypothetical protein 72589 served as an internal standard for *ras2-1* and *tar* normalization, with TaqMan® probe 5′[6-FAM] AG TGG TTG TTA ACA GCG [Tamra-Q]3′, forward primer 5′-AGG AAT TGA TCT CGA AAA ATC TGA A-3′, and reverse primer 5′-GAT CCC ACG CAG AGA AGC AT-3′. This gene was not affected by the presence/absence of endobacteria and displayed a high level of constitutive expression in the RNA-seq data. *Mc* actin gene (105861) was used for normalization in *Mc ras2-1* reactions with TaqMan® probe 5′[6-FAM] CCG AAG TGC AAC TGT TCT TGC CTC ACT [Tamra-Q]3′, forward primer 5′-GCA GGA ATC ACA AAA CGT ATC AAG-3′, and reverse primer 5′-GTT GTG TAT CGC CTG CAT TCT C-3′.

**Identification of sex-relevant genes across fungi.** To identify sex-related genes conserved across fungi, we queried the genomes of *Saccharomyces cerevisiae* YJM789[39], *Schizosaccharomyces pombe* 972h[40], *Aspergillus nidulans* FGSC A4[41], *Cochliobolus heterostrophus* C4[42], *Neurospora crassa* OR74A (N150)[43], *Ustilago maydis* 521[44], *Coprinopsis cinerea* strain Okayama 7 #130[45], *Cryptococcus neoformans* var. *neoformans* B-3501A[46], *Batrachochytrium dendrobatidis* JAM81 (genome.jgi.doe.gov), *Mucor circinelloides* CBS 277.49[47], *Phycomyces blakesleeanus* NRRL1555[47], *R. delemar* 99–880[48], and *Rm* ATCC 52813 (genome.jgi.doe.gov) and ATCC 52814[6]. We additionally included transcriptomic data from *Rhizophagus irregularis* DAOM181602[49]. OrthoMCL[50] was conducted using default parameters (minimum E value cutoff of 1e-5, inflation 1.5) to cluster genes across all of these genomes into orthologous groups. From the genomes of *S. cerevisiae*, *A. nidulans*, *N. crassa*, and *U. maydis*, we extracted a collection of genes with experimentally validated reproductive phenotypes as a tool for further characterization of the resulting clusters, and to assess whether these known genes were also altered in expression due to reproduction in *Rm*. For Ras phylogeny reconstruction, full-length amino acid sequences were aligned using MUSCLE[51] under default parameters, and the phylogeny was constructed with MrBayes[52] under the mixed amino acid substitution model with Γ rate variation run for 2 million generations. For GPCR phylogeny reconstruction, we extracted protein sequences for the seven-transmembrane domain, aligned them with MUSCLE, and reconstructed phylogeny using FastTree[53] under the WAG substitution model[54] with Γ rate variation.

**Rm genome sequencing, assembly, and annotation.** The *Rm* ATCC 52813 genome was sequenced using the Illumina platform. Two Illumina libraries were sequenced: (1) fragment library with 270-bp insert size in 2 × 150-bp reads, and (2) 5.2-kb-long mate pair in 2 × 100-bp reads. Each fastq file was QC filtered for artifact/process contamination and subsequently assembled with AllPathsLG R41043[55], resulting in 26-Mb assembly in 131 scaffolds and 773 contigs, with an average 143.6× read depth coverage. The assembled genome was annotated using the JGI annotation pipeline[56], which combines several gene prediction and functional annotation methods, and integrates the annotated genome into JGI web-based resource for fungal comparative genomics, MycoCosm (http://genome.jgi.doe.gov/fungi)[57].

Before gene prediction, assembly scaffolds were masked using RepeatMasker[58], Repbase library[59], and the most frequent (>150 times) repeats recognized by RepeatScout[60]. The following combination of gene predictors was run on the masked assembly: (1) ab initio, including Fgenesh[61] and GeneMark[62], (2) homology based, including Fgenesh+[61] and GeneWise[61] seeded by BLASTx alignments against the NCBI NR database, and (3) transcriptome based, including Fgenesh and COMBEST[63]. In addition to protein-coding genes, tRNAs were predicted using tRNAscan-SE[64]. All predicted proteins were functionally annotated using SignalP[65] for signal sequences, TMHMM[66] for transmembrane domains, interProScan[67] for integrated collection of functional and structural protein domains, and protein alignments to NCBI NR, SwissProt[68], KEGG[69] for metabolic pathways, and KOG[70] for eukaryotic clusters of orthologs. InterPro[71] and SwissProt[68] hits were used to map Gene Ontology terms[72]. For each genomic locus, the best representative gene model was selected based on a combination of protein homology and transcriptome support, which resulted in the final set of 10,905 gene models reported in this study. Coverage of gene models by BLAT alignments of transcriptome assemblies to the genome assembly resulted in 85% of models being covered over at least 75% of their length, and 67% of models 100% covered. The complete set (100%) of eukaryotic core genes from the CEGMA dataset[73] was found in *Rm*, indicating a reasonably complete genome assembly.

**Data availability.** The transcriptome data are available at the NCBI GEO database under the accession number GSE57644. The *Rhizopus microsporus* ATCC 52813 Whole Genome Shotgun project has been deposited at DDBJ/EMBL/GenBank under accession JOSV00000000. All other relevant data are available in this article and its Supplementary Information files, or from the corresponding author upon request.

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

## Acknowledgements

We thank J. Spatafora for including *Rm* ATCC 52813 in the 1000 Fungal Genomes project, J. Taylor for permission to analyze the unpublished genome of *Batrachochytrium dendrobatidis* JAM81, A. Collmer and J. Worley for the gift of pBS46:YFP and pBS46: mCherry plasmids, E. Angert, and G. Turgeon for comments on the manuscript. The work at Cornell University was funded by the National Science Foundation grants DEB-0918880, IOS-1261004, and DBI-1263103 to T.E.P., and by the National Institutes of Health grant GM-19629 to Susan A. Henry in support of M.L.G., whereas the work conducted by the U.S. Department of Energy Joint Genome Institute was sponsored by the Office of Science of the U.S. Department of Energy under contract DE-AC02-05CH11231.

## Author contributions

S.J.M., O.A.L., M.L.G., and T.E.P. conceived the experiments and wrote the manuscript. I. V.G. coordinated genome project. S.J.M., O.A.L., M.L.G., N.H.S., C.C.B., R.R., H.S., and T.E.P. performed experiments and analyzed the data. T.E.P. secured funding.

## Additional information

**Competing interests:** The authors declare no competing financial interests.

