## [Peer Review File · Nature Communications]

Reviewers' comments:

Reviewer #1 (Remarks to the Author):

I read this manuscript with interest, but I have serious concerns and many comments and questions. Overall, I think the manuscript is not of sufficient quality to warrant publication in Nature Communications. Most importantly, the theory being explained at the start of the manuscript, and central to the story, is not clear and there is other theory on the role of vertical transmission for the transition from antagonism to mutualism that is not being used. Most importantly, the authors describe a scenario where mutualism evolves from antagonism when the 'symbionts take control over their own vertical transmission'. However, there is important theoretical work, for example by Steve Frank (see Frank, 1996 Proc R Soc B), that argues that mutualisms arise when the host is able to enforce vertical transmission, so in other words, with the host taking control. Also, it has been argued that parasitic associations that are vertically transmitted, such as Wolbachia endosymbionts, over time may become more benign and ultimately mutualistic (or 'addictive', if there is compensatory evolution, when the host compensates for the presence of the parasites). Importantly, the results described do not allow to test those different hypotheses.

Below I outline some more comments.

1. The first words of the abstract 'heritable mutualisms'. I think this is not a correct term. 'Heritable' implies two things: i) something is inherited and ii) something or somebody inherits. In other words one partner is active (he who inherits) and the other is passive (he who is inherited). In the case of a host-symbiont interaction, one could make the case that both partners 'inherit each other' in the case of vertical transmission, but I think this should be explained very carefully then, and certainly not be considered as a given.
2. Line 31-32: 'start out as antagonisms in which parasite transmission is constrained by the host'. As explained above, there are examples, where the initial stage is that parasite transmission is constrained by the parasite itself. There are many examples where Wolbachia influences the host to promote its own vertical transmission. And indeed, there are also cases where Wolbachia has turned into mutualisms (or addiction of the host to its Wolbachia symbionts).
3. Line 41: 'and correlated' 'And that this correlated'.
4. Line 45: 'a prerequisite for the antagonism-mutualism transition'. This is too strong. At most you could argue that vertical transmission facilitates the transition.
5. Line 50 'thrive as soil saprotrophs responsible for food spoilage as well as pathogens of plants and immune-compromised patients'. 'as well as' is confusing, since it might mean that they as soil pathogens are also responsible for pathogens of plants, which is not what you mean.
6. With respect to the results, the authors show that only 40% of the sexual spores contain bacteria, which is very little, given the fact that all natural isolates contain this bacterium. So it needs to be emphasised more strongly that this finding raises important questions than now in lines 101-102.
7. The authors focus on vertical transmission, but do not discuss 'how vertical' vertical transmission is, since they have not checked (or I have not read this) if the bacteria are inherited from one parent only, in other words, if transmission during sexual reproduction is uniparental or not. If it is biparental, there is a horizontal component to transmission as well.
8. Figure 1 b. At first, I did not understand what the C-amp results referred to, until at the very end of the manuscript, where this is explained. If you show a figure, the complete figure must be explained, you cannot wait with part of the figure until the end of the manuscript!
9. Line 116: 'genes that are reproductively relevant' 'genes that are associated with reproduction' or something similar.
10. Line 181: Please correct 'to be have been'
11. The English should be checked throughout.

Reviewer #2 (Remarks to the Author):

This is a meaningful, important and interesting paper presenting surprising new data about bacteria and fungi interactions in the zygomycete fungus *Rhizopus microsporus*. The work provides new insight into mating and bacteria-fungi symbioses.

Some aspects of the data presentation are needed to improve the manuscript:

All the tree figures need to use formal Gene locus IDs or accession numbers. the reduce numbering for *Neurospora* 013616T0 is not appropriate if this is NCU13616T0. Why is N150 the strain used here when the community uses OR74A as the reference and the locus IDs from that strain are what are present in reference databases. Similarly because of locus ID changes (ANXX to ANIG_XX) it should be better spelled out what is the locus ID and prefix for the *Aspergillus* and all other gene locus IDs in the trees.

- trees and alignments should also be submitted to TreeBase.

It is unclear what WGA (e.g. methods lines 218-231) is used at all in this study - it is ostensibly to test for presence of endobacteria but the data described as collected in the methods were not described in any fashion. What aspect of the WGA was needed - was the original extraction not sufficient for PCR amplification? I did not see that data used in anywhere in the manuscript.

Figure 2 conveys limited information - it should be clarified. For example the colors for gene type as listed in the center for the circle (Transcription Factors, Pheromone Cascade, Env Sensing) do not appear in a legible fashion in the figure. Where there are genes (?) in groups that are Ascomycete-specific - I assume by the orange color which has no hits anywhere else, it isn't possible to glean what type of genes these are. Linking this figure to the Table S2 order for example with numbers referring to those genes would be helpful for example.

No statistics seem applied to the comparisons made in Figure S2. Are these different or non-significant?

Why is Figure 3 rooted with the orphan receptor - is it as likely there was a single receptor family in the animal ancestor which diversified to create the families shown?

- accession number for the genes in figure 3 for *Rhizopus delemar* are confusing - all the Broad IDs start with RO3G_XXX and these are simply numbers. The JGI site still lists the taxon as *Rhizopus oryzae* Searching 02411 at the site: [http://genome.jgi.doe.gov/pages/search-for-genes.jsf?organism=Rhi or3](http://genome.jgi.doe.gov/pages/search-for-genes.jsf?organism=Rhi%20or3) does not bring up any genes. It seems important that stable accession numbers (e.g. GenBank protein IDs) or stable LOCUS ids for gene names be used to refer to the sequences in the paper.

-- when I search the protein for Rm ATCC52814 (333266) against the *Rhizopus delemar* genome at FungiDB -the best hit is to RO3G_02412 -- so I am unclear if a transposition of the IDs occurred in analyses or what.

--The % identity of some of the proteins in the C GPCR tree (Fig. 3) - is this still just clustering because these are the only the GPCRs in the tree - not sure how clear the GABA sister relationship is from this presentation. *Allomyces* appears to have many copies of the GPCR family too - does inclusion of any of these clarify the relationships?

Table S1 - kind of confusing what you are trying to show - if no differences then simple text that all pairwise combinations of source bacteria and target fungal strain tried yielded reproduction should be sufficient??

Table S2- "Each treatment was replicated at least 3 times." - and was the same result achieved for each replicate?

The genome size of ATCC 52813 is ~1/2 of that we see in the other strains of *R. microsporus* yet there is no comment or discussion on either the genome assembly size differences and how that relates to the other strains. What is the source of this content differences. Is this attributed to the Illumina-only assembly and repeat collapsing or in fact genome streamlining or lack of large scale duplications seen in other strains?

The methods section on lines 333-334 state "CEGMA dataset was found in *Rm*, indicating a reasonably complete genome annotation"  a) CEGMA is assessing genome assembly completeness not annotation so isn't a measure of if the annotation is complete.

Minor:

- Isn't the ref genome for *N. crassa* per the cited publication strain OR74A / FGSC 2489? Why is it referred to as N150?
- Default parameters for OrthoMCL - Please specify what inflation value and what E-value cutoff for BLASTP. Were the *R. irregularis* transcripts -> converted to proteins included in the clustering? Lines 301-302 referred to "we extracted a collection of genes with experimentally validated reproductive phenotypes" - Table S5 ought to include an accession number or stable gene locus ID - gene names can be difficult to track down. As a supplemental table there is no issue with width of the table needing to be printable in a journal format.
- Lined 304-307 - insufficient detail on phylogeny reconstruction are provided. Is there any trimming performed.

Reviewer #3 (Remarks to the Author):

Mondo et al present an exciting study revealing that endosymbiotic bacteria regulates sexual development of the host Mucorales. In addition, the study identifies key factors for the unique sexual reproduction in Mucorales, including trisporic acid receptor gene TAR and ras signaling gene. This study is extremely intriguing because the endosymbiotic system could provide a tool to unveil underlying genetics in Mucorales. However, there are several points the authors have to consider.

In Mucorales, there are two transcription factors, SexM and SexP, are known to regulate sexual development. The authors did not fully cover how the endosymbiotic bacteria regulates the expression of sexM and sexP in each mating type. There are several studies have been done in other Mucorales how sexP/M genes were expressed during mating of *Phycomyces* and *Mucor*. A RNA-seq result table and supplemental data include sexP among differentially regulated genes but no discussion about it there. In addition, a recent *Mucor* study (PLOS Genetics, 2017) found that a neighboring *rnhA* gene to sexP/M is also differentially regulated during sexual reproduction. Therefore, the authors should cover the expression of sex locus genes by the bacteria.

The authors suggest that this system with *R. microspora* and *Burkholderia* can be used as a model to study sexual reproduction in Mucorales. To make this case strong, the authors need to include other Mucorales to see if the TAR and ras genes are regulated during mating.

qPCR for TAR is required to verify the RNA-seq data.

Line 73, is it possible to image of zygospores with fluorescent *Burkholderia*? How one can ensure that there are bacteria in zygospores or not. Although -80C treatment could remove live hyphae, the authors still need to make sure the all hyphal fragments attached to zygospores are dead.

Fig 3. Probably adding known GPCR's of dikarya to the graph would provide a better relationship of theMucorales GPCR's to the human GABA receptor.

Figure 2, please provide what are the color codes. And comprehensive explanation about the illustration is needed.

Reviewers' comments:

Reviewer #1:

I read this manuscript with interest, but I have serious concerns and many comments and questions. Overall, I think the manuscript is not of sufficient quality to warrant publication in Nature Communications. Most importantly, the theory being explained at the start of the manuscript, and central to the story, is not clear and there is other theory on the role of vertical transmission for the transition from antagonism to mutualism that is not being used. Most importantly, the authors describe a scenario where mutualism evolves from antagonism when the 'symbionts take control over their own vertical transmission'. However, there is important theoretical work, for example by Steve Frank (see Frank, 1996 Proc R Soc B), that argues that mutualisms arise when the host is able to enforce vertical transmission, so in other words, with the host taking control. Also, it has been argued that parasitic associations that are vertically transmitted, such as Wolbachia endosymbionts, over time may become more benign and ultimately mutualistic (or 'addictive', if there is compensatory evolution, when the host compensates for the presence of the parasites). Importantly, the results described do not allow to test those different hypotheses.

Indeed, several models exist that describe evolution of mutualisms from antagonisms. In fact, it has been postulated (Aanen & Hoekstra 2007 TREE 22: 506-509), and we agree with this assertion (Lastovetsky et al 2016 PNAS 113: 15102-15107), that the *Rhizopus-Burkholderia* mutualism can be explained by the addiction model. Moreover, because the *Rhizopus* host is addicted to its *Burkholderia* endobacteria for asexual reproduction, the model that we are exploring (symbiont control over host reproduction) and the addiction model are not mutually exclusive. As for the Frank's model of host control over symbiont transmission, we believe that it applies to systems in which symbionts already have net beneficial effects on the host rather than describing circumstances that facilitate transitions from antagonism to mutualism, which our paper is focused on. Intriguingly, our preliminary population data suggest that *Burkholderia* endobacteria are transmitted uniparentally despite the fact that their hosts mate and bacteria should be able to mix. It cannot be excluded, therefore, that all three models apply to our study system.

In the present version, we clarified distinctions between these three models as follows:

Lines 69-72: "Such host dependence on endosymbionts for asexual reproduction is consistent with the addiction model of mutualism evolution^{3,7}. According to this model, a host population that interacts with an antagonistic symbiont should develop mechanisms to compensate for its negative effects and become addicted to the symbiont's continued presence⁸."

Lines 119-125: "These findings indicate that sufficient control has been achieved by the endosymbiont to facilitate its own vertical transmission, allowing the antagonism-mutualism

transition in heritable symbioses, in which symbiont propagation is initially controlled by the host². Evolutionary theory predicts further that once a mutualism is established, the host is favored to control mixing of symbionts and thereby reduce their harmful competition for the host resources¹⁴. It remains to be investigated whether such control is in place in the *Rhizopus-Burkholderia* symbiosis.”

Below I outline some more comments.

1. *The first words of the abstract ‘heritable mutualisms’. I think this is not a correct term. ‘Heritable’ implies two things: i) something is inherited and ii) something or somebody inherits. In other words one partner is active (he who inherits) and the other is passive (he who is inherited). In the case of a host-symbiont interaction, one could make the case that both partners ‘inherit each other’ in the case of vertical transmission, but I think this should be explained very carefully then, and certainly not be considered as a given.*

We appreciate Reviewer’s concern about our use of the phrase “heritable mutualisms”. In response, we revised the sentence to read: “Heritable mutualisms, in which beneficial symbionts are transmitted vertically between generations of the host.” The phrase “heritable mutualism” is an expansion of a term “heritable symbiosis”, which is commonly used to describe vertically transmitted endobacteria of insects (Bennett & Moran 2015 *PNAS* **112**, 10169-10176).

2. *Line 31-32: ‘start out as antagonisms in which parasite transmission is constrained by the host’. As explained above, there are examples, where the initial stage is that parasite transmission is constrained by the parasite itself. There are many examples where Wolbachia influences the host to promote its own vertical transmission. And indeed, there are also cases where Wolbachia has turned into mutualisms (or addiction of the host to its Wolbachia symbionts).*

Again, we appreciate this comment. Our statement “Evolutionary theory suggests that many heritable mutualisms originate as antagonisms with parasite transmission constrained by the host” is not intended to indicate that ALL mutualism originate from antagonisms with host control over symbiont transmission. We are simply setting up the stage for the model that we explore in the paper.

3. *Line 41: ‘and correlated’ ‘And that this correlated’.*

We changed to: “We found that removal of endobacteria eliminated or reduced mating in fungi, and was correlated with *ras2* expression.”

4. *Line 45: ‘a prerequisite for the antagonism-mutualism transition’. This is too strong. At most you could argue that vertical transmission facilitates the transition.*

We revised as follows: “Our results demonstrate that regulating one of the host reproductive modes (asexual proliferation) and partial control over the other (sexual reproduction) are sufficient for symbiont’s control of its vertical transmission. Such control is expected to facilitate an antagonism-to-mutualism transition in heritable symbioses that start out with host control of symbiont propagation.” (lines 46-50).

5. *Line 50 ‘thrive as soil saprotrophs responsible for food spoilage as well as pathogens of plants and immune-compromised patients’. ‘as well as’ is confusing, since it might mean that they as soil pathogens are also responsible for pathogens of plants, which is not what you mean.*

We revised as follows: “Like many other Mucoromycotina, the *Rm* hosts of *Burkholderia* thrive as soil saprotrophs. They can cause food spoilage, infect plants⁵, and act as opportunistic pathogens of immune-compromised humans⁶.”

6. *With respect to the results, the authors show that only 40% of the sexual spores contain bacteria, which is very little, given the fact that all natural isolates contain this bacterium. So it needs to be emphasised more strongly that this finding raises important questions than now in lines 101-102.*

We revised as follows: “Importantly, less than half of zygospores formed during mating interactions of wild hosts harbors endobacteria, and it remains untested whether zygospore functionality is affected by the endosymbiont presence.”

7. The authors focus on vertical transmission, but do not discuss 'how vertical' vertical transmission is, since they have not checked (or I have not read this) if the bacteria are inherited from one parent only, in other words, if transmission during sexual reproduction is uniparental or not. If it is biparental, there is a horizontal component to transmission as well.

As we indicated above, our preliminary data on population structure of *Burkholderia* endobacteria suggest lack of recombination, even though their fungal hosts are capable of mating. This observation suggests that endosymbiont transmission is uniparental in the *Rhizopus-Burkholderia* symbiosis. These data will be presented in a separate manuscript.

8. Figure 1 b. At first, I did not understand what the C-amp results referred to, until at the very end of the manuscript, where this is explained. If you show a figure, the complete figure must be explained, you cannot wait with part of the figure until the end of the manuscript!

Panel b is included in Figure 1 because it illustrates that cured fungi can still mate at a low rate. We refer to it in line 92, when we discuss incomplete control of mating. To clarify, we revised figure caption to read: "Effects of endobacteria and exogenous di-buteryl cAMP on the formation of zygospores during mating between ATCC 52813 and ATCC 52814 with cured strains showing restricted mating".

9. Line 116: 'genes that are reproductively relevant' 'genes that are associated with reproduction' or something similar.

As suggested, we rephrased to read: "Because the knowledge of molecular mechanisms that underlie sexual reproduction in Mucoromycotina lags behind other fungi, we first needed to identify genes that are relevant to reproductive processes in these fungi."

10. Line 181: Please correct 'to be have been'
Thank you! Done!

11. The English should be checked throughout.
The narrative has been read carefully by multiple readers.

Reviewer #2:

*This is a meaningful, important and interesting paper presenting surprising new data about bacteria and fungi interactions in the zygomycete fungus *Rhizopus microsporus*. The work provides new insight into mating and bacteria-fungi symbioses.*

Some aspects of the data presentation are needed to improve the manuscript:

*All the tree figures need to use formal Gene locus IDs or accession numbers. the reduce numbering for *Neurospora* 013616T0 is not appropriate if this is NCU13616T0. Why is N150 the strain used here when the community uses OR74A as the reference and the locus IDs from that strain are what are present in reference databases. Similarly because of locus ID changes (ANXX to ANIG_XX) it should be better spelled out what is the locus ID and prefix for the *Aspergillus* and all other gene locus IDs in the trees. We agree that stable gene locus IDs are necessary, and, therefore, have added to all tree figures a stable locus ID in the form of GenBank accession number or JGI protein ID following species and strain identification. N150 and OR74A are synonyms (Galagan et al 2003 Nature 422, 859-868), but we agree that OR74A should be emphasized, as it is most commonly used identifier for this strain.*

- trees and alignments should also be submitted to TreeBase.

We found TreeBase to be unreliable. Therefore, we include alignments and trees as datasets S3, S4 and S5 in SI.

It is unclear what WGA (e.g methods lines 218-231) is used at all in this study - it is ostensibly to test for presence of endobacteria but the data described as collected in the methods were not described in any fashion. What aspect of the WGA was needed - was the original extraction not sufficient for PCR amplification? I did not see that data used in anywhere in the manuscript.

WGA was used on DNA from individually dissected zygosporangia to allow detection of endobacteria by PCR targeting *Burkholderia* 23 rRNA gene sequences. The WGA step was needed to generate template DNA for multiple PCR reactions per zygosporangium to safeguard from experimental failures, which we now explain in lines 260-263. The results of this experiment are presented in lines 82-85.

Figure 2 conveys limited information - it should be clarified. For example the colors for gene type as listed in the center for the circle (Transcription Factors, Pheromone Cascade, Env Sensing) do not appear in a legible fashion in the figure. Where there are genes (?) in groups that are Ascomycete-specific - I assume by the orange color which has no hits anywhere else, it isn't possible to glean what type of genes these are. Linking this figure to the Table S2 order for example with numbers referring to those genes would be helpful for example.

As requested, we have substantially revised Figure 2. It is now divided into three panels: (a) all OrthoMCL gene clusters, (b) genes DE during sexual reproduction of *R. microsporus*, and (c) regulators of sexual reproduction. Genomes representing major fungal lineages are marked with distinct colors. Regulators of sexual reproduction (pheromone MAPK cascade, transcription factors and environmental sensors) are specifically identified in panel (c). Lastly, OrthoMCL data used to generate all data in Figure 2 are now available, ordered as shown in Fig 2a, displayed in Dataset S2.

No statistics seem applied to the comparisons made in Figure S2. Are these different or non-significant? As requested, we provided statistical analysis of colony expansion in Figure S2.

Why is Figure 3 rooted with the orphan receptor - is it as likely there was a single receptor family in the animal ancestor which diversified to create the families shown?

A similar suggestion was made by Reviewer 3. In response, we have revised Figure 3 substantially. We collected sequences from GPCRdb across 4 different classes (class A, B1, B2, and F), as well as class D GPCRs from Dikarya (as per Reviewer 3 suggestion) and built a phylogeny using the 7-transmembrane domain of these GPCRs. The resulting phylogeny shows that GPCRs identified in this study are nested within class C, and very distant from class D found in other fungi.

- accession number for the genes in figure 3 for *Rhizopus delemar* are confusing - all the Broad IDs start with RO3G_XXX and these are simply numbers. The JGI site still lists the taxon as *Rhizopus oryzae* Searching 02411 at the site: <http://genome.jgi.doe.gov/pages/search-for-genes.jsf?organism=Rhior3> does not bring up any genes. It seems important that stable accession numbers (e.g. GenBank protein IDs) or stable LOCUS ids for gene names be used to refer to the sequences in the paper.

We have now updated all trees to show stable locus IDs, as discussed above.

-- when I search the protein for *Rm* ATCC52814 (333266) against the *Rhizopus delemar* genome at FungiDB -the best hit is to RO3G_02412 -- so I am unclear if a transposition of the IDs occurred in analyses or what.

Thank you for catching this! It appears that in the JGI *R. delemar* database a transposition issue did occur, as the model RO3G_02412 from fungiDB (and in Genbank) is listed as RO3G_02411 in MycoCosm. For this reason, we have changed all *R. delemar* gene models to numbers from GenBank and provide GenBank accession numbers now.

--The % identity of some of the proteins in the C GPCR tree (Fig. 3) - is this still just clustering because these are the only the GPCRs in the tree - not sure how clear the GABA sister relationship is from this presentation. *Allomyces* appears to have many copies of the GPCR family too - does inclusion of any of these clarify the relationships?

To improve clarity, we have expanded the GPCR tree substantially, as described above. Instead of using *Allomyces*, we included *Catenaria anguillulae*, as we noticed that almost the entire genome of *Allomyces* was duplicated (therefore doubling gene content).

Table S1 - kind of confusing what you are trying to show - if no differences then simple text that all pairwise combinations of source bacteria and target fungal strain tried yielded reproduction should be

sufficient??

As suggested, we removed Table S1 and explained it in the text: "Reinfection with endobacteria restored both asexual and sexual reproduction regardless of whether bacteria were introduced into their original native hosts or non-native hosts, with all pairwise combinations of source bacteria and target fungal strains across ATCC 52811, ATCC 52813, ATCC 52814, and ATCC 62417 yielding reproduction." (lines 99-103).

Table S2- "Each treatment was replicated at least 3 times." - and was the same result achieved for each replicate?

To clarify, we rephrased: "Each observation summarizes three replicate cultures."

The genome size of ATCC 52813 is ~1/2 of that we see in the other strains of *R. microsporus* yet there is no comment or discussion on either the genome assembly size differences and how that relates to the other strains. What is the source of this content differences. Is this attributed to the Illumina-only assembly and repeat collapsing or in fact genome streamlining or lack of large scale duplications seen in other strains? Indeed, the genome of ATCC 52813 is smaller than the genomes of *R. delemar* 99-880 (previously referred to as *R. oryzae* 99-880) and *R. chinensis* CCTCCM201021 (previously referred to as *R. microsporus* CCTCC M201021). The genome of *R. delemar* is thought to have undergone a whole genome duplication event followed by gene loss (Ma et al 2009 PLoS Genet 5, e1000549), which may explain its size expansion compared to the genomes of *R. microsporus* ATCC 52813, ATCC 52814, ATCC 11559 which are 24 to 26 Mb in size. We added this information to Table S3, which summarizes genome assembly statistics for *Rm* ATCC 52813 and comparative organisms from the Mucoromycotina subphylum.

The methods section on lines 333-334 state "CEGMA dataset was found in *Rm*, indicating a reasonably complete genome annotation"  a) CEGMA is assessing genome assembly completeness not annotation so isn't a measure of if the annotation is complete.

We changed this to read: "indicating a reasonably complete genome assembly" (lines 396-398).

Minor:

- Isn't the ref genome for *N. crassa* per the cited publication strain OR74A / FGSC 2489? Why is it referred to as N150?

As per Galagan et al 2003 (Nature 422, 859-868) these are synonyms; we added OR74A for clarity.

- Default parameters for OrthoMCL - Please specify what inflation value and what E-value cutoff for BLASTP. Were the *R.irregularis* transcripts -> converted to proteins included in the clustering?

We changed the text as follows: "OrthoMCL⁴⁷ was conducted using default parameters (minimum E value cutoff of 1e-5, inflation 2) to cluster genes across all of these genomes into orthologous groups."

Yes, *R. irregularis* transcripts were converted to proteins prior to clustering.

Lines 301-302 referred to "we extracted a collection of genes with experimentally validated reproductive phenotypes" - Table S5 ought to include an accession number or stable gene locus ID - gene names can be difficult to track down. As a supplemental table there is no issue with width of the table needing to be printable in a journal format.

We have now added this information to Table S5.

- Lined 304-307 - insufficient detail on phylogeny reconstruction are provided. Is there any trimming performed.

No trimming was performed for these alignments. We expanded details in our methods section regarding phylogeny reconstruction to read: "For Ras phylogeny reconstruction, full-length amino acid sequences were aligned using MUSCLE⁴⁸ under default parameters, and the phylogeny was constructed using MrBayes⁴⁹ under the mixed amino acid substitution model with Γ rate variation run for 2 million generations. For GPCR phylogeny reconstruction, we extracted protein sequences for the 7-transmembrane domain, aligned them with MUSCLE, and

reconstructed phylogeny using FastTree⁵⁰ under the WAG substitution model⁵¹ with Γ rate variation." (lines 365-371).

Reviewer #3:

Mondo et al present an exciting study revealing that endosymbiotic bacteria regulates sexual development of the host Mucorales. In addition, the study identifies key factors for the unique sexual reproduction in Mucorales, including trisporic acid receptor gene TAR and ras signaling gene. This study is extremely intriguing because the endosymbiotic system could provide a tool to unveil underlying genetics in Mucorales. However, there are several points the authors have to consider.

In Mucorales, there are two transcription factors, SexM and SexP, are known to regulate sexual development. The authors did not fully cover how the endosymbiotic bacteria regulates the expression of sexM and sexP in each mating type. There are several studies have been done in other Mucorales how sexP/M genes were expressed during mating of Phycomyces and Mucor. A RNA-seq result table and supplemental data include sexP among differentially regulated genes but no discussion about it there. In addition, a recent Mucor study (PLOS Genetics, 2017) found that a neighboring rhnA gene to sexP/M is also differentially regulated during sexual reproduction. Therefore, the authors should cover the expression of sex locus genes by the bacteria.

Thank you for this suggestion! While, previously, we did not mention sex loci, as endosymbionts had no effect on their expression levels, we realize that their importance during reproduction warrants a more extensive discussion. We now comment as follows: "As expected, we also observed significant upregulation of mating type genes, *sexM* and *sexP*, previously discovered in *Mucoromycotina*¹⁷." (lines 147-148). Additionally, as recommended, we looked into expression of *rhnA*, which in both mates showed no upregulation during sexual reproduction or due to endosymbionts: "In contrast, the neighboring *rhnA* gene¹⁸ was not differentially expressed during reproduction in *Rm*." (lines 149-150).

The authors suggest that this system with R. microspora and Burkholderia can be used as a model to study sexual reproduction in Mucorales. To make this case strong, the authors need to include other Mucorales to see if the TAR and ras genes are regulated during mating.

We now include qRT-PCR data on *ras2-1* expression in *Mucor circionelloides* (Fig 4b). These data suggest that *ras2-1* is upregulated during sexual reproduction of *Mc* relative to vegetative growth. We are also in the process of functional characterization of candidate *tar* genes in *Mc*, which will be presented in a separate manuscript.

qPCR for TAR is required to verify the RNA-seq data.

As requested, we generated qRT-PCR data for the candidate *tar* genes in *Rm* using a partially different pair of mates from the one that was used for RNAseq. These data are now included in Figure 3b and c.

Line 73, is it possible to image of zygospores with fluorescent Burkholderia? How one can ensure that there are bacteria in zygospores or not. Although -80C treatment could remove live hyphae, the authors still need to make sure the all hyphal fragments attached to zygospores are dead.

We tried to image mCherry- and YFP-labeled endobacteria in zygospores after crushing them. Unfortunately, autofluorescence of the zygospore wall obscures fluorescent signals of mCherry and YFP markers. As for killing hyphae attached to zygospores, we exposed them 10% w/v chloramine T for 20 min, a treatment that we found sufficient to kill hyphae of *Rm* (lines 257-259).

Fig 3. Probably adding known GPCR's of dikarya to the graph would provide a better relationship of the Mucorales GPCR's to the human GABA receptor.

A similar suggestion was made by Reviewer 2. In response, we have considerably reworked this phylogeny. It now includes class D GPCRs, more taxa of early-diverging fungi, as well as GPCRs across multiple classes.

Figure 2, please provide what are the color codes. And comprehensive explanation about the illustration is needed.

As requested, we have substantially revised Figure 2. It is now divided into three panels: (a) all OrthoMCL gene clusters, (b) genes DE during sexual reproduction of *R. microsporus*, and (c) regulators of sexual reproduction. Genomes representing major fungal lineages are marked with distinct colors. Regulators of sexual reproduction (pheromone MAPK cascade, transcription factors and environmental sensors) are specifically identified in panel (c). Lastly, all data in Figure 2 are displayed in Dataset S2.

REVIEWERS' COMMENTS:

Reviewer #1 (Remarks to the Author):

The authors have done an excellent job in rebutting my concerns. I do not have any further comments.

Reviewer #2 (Remarks to the Author):

I'm mostly comfortable with the current response to the points raised in my prior review.

Reviewer #3 (Remarks to the Author):

The authors addressed the concerns raised by three reviewers well and accordingly revised the manuscript substantially. In my opinion, the ms reads well to provide a new finding to the field.

Minor concerns about Fig 3. For the red line for Class C (others), does it include GABA? What's 'others'? The figure legend does not have explanation for the Class C (others). And for the green box, Class C (Dikarya) must be Class 'D' (Dikarya).
Some descriptions for the Class B1, B2, F, and A would be helpful too.

We very much appreciate comments and suggestions from the Reviewers on our manuscript "**Bacterial endosymbionts influence host sexuality and reveal reproductive genes of early divergent fungi**" (NCOMMS-17-04445). Below, we elaborate on how we revised the manuscript in response to them.

Reviewers' comments:

Reviewer #1 (Remarks to the Author):

The authors have done an excellent job in rebutting my concerns. I do not have any further comments.

Reviewer #2 (Remarks to the Author):

I'm mostly comfortable with the current response to the points raised in my prior review.

Reviewer #3 (Remarks to the Author):

The authors addressed the concerns raised by three reviewers well and accordingly revised the manuscript substantially. In my opinion, the ms reads well to provide a new finding to the field.

Minor concerns about Fig 3. For the red line for Class C (others), does it include GABA? What's 'others'? The figure legend does not have explanation for the Class C (others). And for the green box, Class C

(Dikarya) must be Class 'D' (Dikarya).

Some descriptions for the Class B1, B2, F, and A would be helpful too.

As suggested, we added protein names the Class C GPCRs, which allows for rapid identification of GABA receptors. We also included brief descriptors of other GPCR classes.